# Brain-wide connectome inferences using functional connectivity MultiVariate Pattern Analyses (fc-MVPA)

**Alfonso Nieto-Castanon**[1,2]*

**1** Department of Speech, Language, and Hearing Sciences, Boston University, Boston, Massachusetts, United States of America, **2** Department of Brain and Cognitive Sciences, Massachusetts Institute of Technology, Cambridge, Massachusetts, United States of America

* alfnie@bu.edu

## Abstract

Current functional Magnetic Resonance Imaging technology is able to resolve billions of individual functional connections characterizing the human connectome. Classical statistical inferential procedures attempting to make valid inferences across this many measures from a reduced set of observations and from a limited number of subjects can be severely underpowered for any but the largest effect sizes. This manuscript discusses fc-MVPA (functional connectivity Multivariate Pattern Analysis), a novel method using multivariate pattern analysis techniques in the context of brain-wide connectome inferences. The theory behind fc-MVPA is presented, and several of its key concepts are illustrated through examples from a publicly available resting state dataset, including an analysis of gender differences across the entire functional connectome. Finally, Monte Carlo simulations are used to demonstrate the validity and sensitivity of this method. In addition to offering powerful whole-brain inferences, fc-MVPA also provides a meaningful characterization of the heterogeneity in functional connectivity across subjects.

## Author summary

The human connectome comprises billions of functional connections between distant brain areas. In recent years, analyses of functional Magnetic Resonance Imaging (fMRI) data have provided large amounts of information exploring the differences in the human connectome across individuals, developmental trajectories, or mental states. However, scientists' ability to derive strong conclusions from the analysis of these data are often hindered by the sheer number of connections analyzed, where only connections that show exceptionally large effects are able to stand out against that vast background. This leads to results that tend to overemphasize similarities and mask out differences that are either weaker or distributed across multiple individual connections, potentially misleading conceptual models of the human connectome. This manuscript discusses a novel method for the analysis of the human connectome (functional connectivity Multivariate Pattern Analysis) that addresses these limitations and enables strong conclusions from fMRI data by

**Data Availability Statement:** This manuscript analyses are based on a publicly available resting state dataset (Cambridge 1000-connectomes dataset, n=198; www.nitrc.org/projects/fcon_1000; doi:10.1073/pnas.0911855107), and all methods

described in this manuscript are publicly available as part of CONN (RRID:SCR_009550, www.nitrc.org/projects/conn).

**Funding:** This research was supported by the National Institute on Deafness and Other Communication Disorders (R01 DC007683, R01 DC002852, R01 DC016270), National Institute of Neurological Disorders and Stroke (U01 NS117836) and National Institute of Mental Health (U01 MH108168).

**Competing interests:** The authors have declared that no competing interests exist.

combining classical statistics with modern pattern analysis techniques. This technique is exemplified using a publicly available database of resting state data to characterize some of the main aspects of the human connectome that differ across individuals, and to identify specific differences in the human connectome across gender.

This is a *PLOS Computational Biology* Methods paper.

## Introduction

Functional connectivity Magnetic Resonance Imaging (fcMRI) is used to characterize the state (e.g. during rest or during individual cognitive tasks) of the human connectome, the set of all functional connections within the brain. In its most basic form, the entire human connectome state can be represented in a way that is limited only by the spatial resolution of the MRI acquisition sequence as a matrix of voxel-to-voxel functional connectivity values. Human connectome research is often motivated by the attempt to characterize similarities and discrepancies in these functional connectivity matrices across subjects or across experimental conditions, performing inferences that extrapolate from the limited data available in a study sample to properties of the human connectome in a larger population. However, this form of unconstrained **brain-wide connectome inferences** can suffer from a curse of dimensionality. A mass-univariate approach analyzing each functional connection between every pair of voxels in the brain may consist of over 60 *billion* individual statistical tests (the total number of pairwise functional connections among approximately 250,000 isotropic 2mm voxels within gray matter areas). This poses considerable difficulties. First, analytically, as an appropriate correction for multiple comparisons across this abundance of tests results in exceedingly low sensitivity. For example, simple Bonferroni or False Discovery Rate corrections [1,2] would require at least one individual connection below a $p < 10^{-12}$ significance level in order to resolve FWE-corrected significance at the analysis level, limiting the sensitivity and applicability of these analyses (e.g. [3]). Second, computationally, as each voxel-to-voxel correlation matrix would require approximately 400Gb of memory or storage space for each individual subject and experimental condition of interest, making storing and working with these matrices extraordinarily demanding. Third, practically, as the potential wealth of information of voxel-to-voxel connectivity results makes reporting and interpreting the results of these analyses a significant challenge in itself.

Existing approaches have addressed these issues by either narrowing the focus of the analyses to connectivity with one or a few a priori seed areas (e.g. connectivity with amygdala) and then performing seed-based connectivity analyses (SBC), or by limiting the analysis units from voxels to larger parcels or Regions of Interest (ROIs) and then performing ROI-to-ROI connectivity analyses (RRC). The first approach (SBC) reduces the multiple-comparison problem by focusing on individual (or linear combinations of) rows of the functional connectivity matrices, disregarding all other possible effects beyond those involving at least one of these a priori seed areas. The main advantage of this approach is its simplicity, as it can take advantage of the same cluster-level inferential procedures that have been proven effective in standard analyses of functional activation, such as Gaussian Random Field theory inferences [4], permutation/randomization analyses [5] or Threshold Free Cluster Enhancement (TFCE) [6]. The main disadvantage of this approach is a high chance of potential false negatives, as other effects not involving the chosen seed areas may be missed.

The second approach (RRC) is able to perform brain-wide connectome inferences with sufficient sensitivity by drastically reducing the multiple-comparison problem, typically focusing on no more than a few hundred ROIs, often defining an entire parcellation of the brain (e.g. Harvard-Oxford atlas). In addition, specialized false positive control approaches, such as Functional Network Connectivity [7], Network Based Statistics [8], Spatial Pairwise Clustering [9], Threshold Free Cluster Enhancement [6], or Multivariate cNBS [10] can be used to further increase the sensitivity of these analyses. Nevertheless, ROI-to-ROI analyses suffer from reduced spatial specificity arising from the a priori selection of parcels of interest, and their conclusions can be particularly sensitive to the choice of ROIs. Advances in connectivity-based parcellations (e.g. [11]) or subject-specific functional ROIs (e.g. [12]) can be useful to partially alleviate these concerns.

An alternative approach uses Principal Component Analyses (PCA) or Independent Component Analyses (ICA) to assess differences in functional networks, or sets of functionally correlated areas, across groups [13]. Similar to seed-based approaches, PCA and ICA are able to drastically reduce the multiple-comparison problem by focusing on individual networks, each comprising a group of functional-related areas, and then evaluating measures of within- and between- network connectivity. One advantage of this approach, compared to SBC, is that these networks do not need to be defined a priori and can be instead estimated directly from the functional data. In the context of brain-wide connectome inferences, nevertheless, these methods suffer from similar shortcomings as SBC, namely the potential of false negatives, where finer functional connectivity differences that are not present at the level of entire networks may be missed.

This paper proposes functional connectivity Multivariate Pattern Analysis (fc-MVPA), an alternative approach to the analysis of the brain-wide connectome at the resolution of individual voxels, that overcomes the difficulties of brain-wide connectome analyses using multivariate pattern analysis techniques. Like other MVPA approaches in neuroimaging, fc-MVPA follows a general searchlight procedure (e.g. [14]), but instead of focusing on the pattern of activation surrounding each voxel, fc-MVPA considers separately for each voxel the entire multivariate pattern of functional connections between this voxel and the rest of the brain. Most MVPA methods focus on the relationship between properties of these multivariate patterns, characterizing a subject's mental state, and either static (e.g. patients vs. controls) or dynamic properties of the experimental design (e.g. pre- vs. after- intervention, task vs. rest, etc.), while varying in how exactly these relationships are modeled or analyzed. Classical MVPA analyses (e.g. [15,16]) attempt to estimate, from these or other searchlight patterns, properties of the experimental paradigm. These MVPA analyses are often referred to as ***backward models*** [17], and typically use machine learning classification models embedded in a cross-validation framework to decode information about a subject's mental state from their activation patterns. In this popular class of MVPA models searchlight patterns act as independent/explanatory variables, while known subject or experimental properties act as dependent/outcome variables. Unlike those forms of MVPA, fc-MVPA instead uses a ***forward model*** of the data, attempting to estimate the shape of these searchlight patterns from known subject or experimental properties, switching the role of dependent/independent variables. In addition to being advantageous in terms of the interpretability of model weights, forward models also enable us to frame brain-wide connectome inferences in the context of the General Linear Model (GLM), one of the most widely used inferential statistical methods in neuroimaging, and use powerful multivariate inferences [10] to directly address researchers' hypotheses. Specifically, this approach allows us to make statistical inferences about individual voxels in the brain regarding the *shape* of their functional connectivity patterns (e.g. is the shape of the functional connectivity pattern between a voxel and the rest of the brain different in patient vs.

control subjects?), and then repeat these analyses across all voxels within the brain extending these inferences to the entire connectome (e.g. is the shape of functional connectivity patterns different in patient vs. control subjects anywhere in the brain?).

Code implementing fc-MVPA, as described in this manuscript, is available in SPM's CONN functional connectivity toolbox [18,19] and it has already been evaluated and shown to provide valuable insights on a variety of research topics (e.g. [20–45]). Despite this, a rigorous discussion of this method's approach, validity, and applications has been missing. This manuscript aims to correct that record by presenting a detailed mathematical description of the fc-MVPA method, highlighting its relationship to other multivariate statistical inferential procedures, illustrating some of fc-MVPA key concepts and applications through examples, and demonstrating the method's validity and sensitivity through Monte Carlo simulations.

In the first section, a general framework to perform univariate and multivariate statistical inferences in the context of functional connectivity data is presented. This is followed by a section describing the specifics of the fc-MVPA approach, where some of its central applications, including brain-wide connectome inferences as well as the characterization of intersubject heterogeneity, are further illustrated with examples. Finally, the last section presents simulation results demonstrating the method's validity, and discussing some of the main factors affecting sensitivity. All analysis examples in this manuscript are based on a publicly available resting state dataset (Cambridge 1000-connectomes dataset, n = 198; www.nitrc.org/projects/fcon_1000), and all methods are publicly available in SPM12 [46] (RRID:SCR_007037, www.fil.ion.ucl.ac.uk/spm/software/spm12/) and CONN [19] (RRID:SCR_009550, www.nitrc.org/projects/conn).

## Methods

### Definition of functional-connectivity univariate and multivariate analyses (fc-MUA, fc-MVA)

For any subject $n$ in a study, and any pair of voxels $x$ and $y$, we consider the values $r_n(x,y)$ characterizing the functional connectivity between these two voxels for this subject. Without loss of generality, we are going to consider $r_n(x,y)$ to represent Pearson correlation coefficients between these two voxels BOLD timeseries (but all of the following descriptions would equally apply to any arbitrary connectivity or similarity measure between pairs of elements in any high-dimensional set). In a study we desire to make inferences regarding the properties of these connectivity measures in the population from which the study subjects are being drawn. For example, we may ask, based on our current study data, whether there are any connectivity differences when comparing patients to control subjects, or whether functional connectivity strength correlates with age, or whether it is modulated by some experimental condition. To that end we could use a forward model of the data defining for each individual connection a separate General Linear Model (GLM) of the form:

functional connectivity Mass-Univariate Analyses (fc-MUA)

$$\forall x, y \qquad r_n(x, y) = \boldsymbol{g}_n \cdot \boldsymbol{b}(x, y) + \boldsymbol{\varepsilon}_n(x, y) \cdot \sigma(x, y)$$
$$\text{Null hypothesis}: \qquad \boldsymbol{C} \cdot \boldsymbol{b}(x, y) = 0 \tag{1}$$

(note on notation: in this document's equations we use regular fonts to refer to scalars, bolded lower case fonts for vectors, and bolded capital fonts for matrices; see glossary of terms in the supplementary materials S1 Table for details about the interpretation and dimensionality of all terms in these equations.) In Eq (1) $\boldsymbol{g}_n$ is a predictor vector for each subject $n$

characterizing known factors in our experimental design, such as group membership or behavioral measures (also known as the model *design matrix*), $b(x,y)$ is an unknown vector of *regression coefficients*, estimated from the data and characterizing the effect of each modeled predictor on the outcome functional connectivity measures (e.g. the average connectivity strength within each group), $\varepsilon_n(x,y)$ represents an error term sampled independently for each subject from a random Gaussian field with zero mean and unit variance (GLM asymptotic normality assumption), and $\sigma(x,y)$ is an intersubject variance term that depends on position $x$ but is otherwise constant across subjects (GLM homoscedasticity assumption). The General Linear Model uses Ordinary Least Squares (OLS) to computed an unbiased estimate of the vector $b$ from the data (Gauss-Markov theorem). After estimating these regressor coefficients, we can specify a null-hypothesis of the form $C \cdot b(x,y) = 0$ for any given between-subjects contrast $C$ (e.g. a null hypothesis might evaluate whether functional connectivity differs between patients and controls) and use a classical hypothesis testing framework to evaluate this hypothesis. Hypothesis testing extrapolates from the observed properties of $b$, estimated only from our sample data, to the effects of the associated predictor terms in the larger population, allowing us to make valid inferences about any hypothesis in this larger population. Null-hypotheses are generally evaluated using a Likelihood Ratio Test (LRT) based on a Wilks Lambda distribution and associated $T$- and $F$- statistics [47]. These hypotheses are tested separately for each pair of voxels $x$ and $y$, resulting in a statistical parametric map $F(x,y)$ of $T$- or $F$- statistics and associated p- values, characterizing the likelihood of our observations under the null hypothesis for every individual seed and target voxels.

We refer to this approach as *mass-univariate* (fc-MUA) because it is based on a separate univariate test for each connection (for each pair of voxels $x$ and $y$) in the entire brain-wide connectome. As mentioned before, one of the main difficulties with fc-MUA in the context of brain-wide connectome inferences is the extremely large number of connections evaluated (one for every pair of voxels) leading to the need for very strong multiple comparison corrections and reduced sensitivity to detect anything but the largest effects.

Compared to this mass-univariate approach, functional connectivity MultiVariate Analyses (fc-MVA) use a searchlight approach where each individual analysis focuses on one individual voxel-of-interest $x$, and perform an omnibus test of the connectivity between this voxel and the rest of the brain using a multivariate GLM analysis of the form:

functional connectivity MultiVariate Analyses (fc-MVA)

$$\forall x \qquad r_n(x) = g_n \cdot B(x) + \varepsilon_n(x) \cdot \Sigma(x)$$
$$\text{Null hypothesis}: \qquad C \cdot B(x) \cdot P(x) = 0 \tag{2}$$

The term $r_n(x,y)$ in Eq 2 contains an entire map of connectivity values defined in vector form (each element of this vector contains the connectivity value $r_n(x,y)$ for a different target voxel $y$), fully characterizing the functional connectivity pattern for subject $n$ between the seed-voxel $x$ and the rest of the brain. $B(x)$ is now an unknown predictors-by- voxels matrix of regression coefficients, $\varepsilon_n(x)$ is a residual error vector sampled from a random multivariate Gaussian distribution with zero mean and unit variance, and $\Sigma(x)$ is a voxels-by-voxels semi-positive definite matrix characterizing the spatial covariance in functional connectivity patterns, which again may depend on position $x$ but is otherwise constant across subjects. As before, the General Linear Model uses Ordinary Least Squares (OLS) to estimate the matrix of regressor coefficients $B$ from the data. In the context of hypothesis testing, $C$ and $P(x)$ represent the between-subjects and between-voxels contrast matrices, respectively, characterizing

which aspects of the matrix of regression coefficients we would like to evaluate. Any arbitrary hypothesis of the form $\boldsymbol{C}^t \cdot \boldsymbol{B}(x) \cdot \boldsymbol{P}(x) = \boldsymbol{0}$, may be evaluated separately for each searchlight voxel $x$ using a statistical parametric map $F(x)$, computed using a Satterthwaite approximation [48]:

$$F(x) = \frac{b}{c} \frac{tr(\boldsymbol{H}(x))}{tr(\boldsymbol{W}(x))} \sim F_{kc,kb} \tag{3}$$

$$b \equiv N - rank(\boldsymbol{G})$$

$$c \equiv rank(\boldsymbol{G}\boldsymbol{C}^t)$$

$$k \equiv \frac{tr(\boldsymbol{W}(x))^2}{tr(\boldsymbol{W}^2(x))}$$

where $b$ and $c$ are error and hypothesis degrees of freedom, respectively, and the matrices $\boldsymbol{W}$ and $\boldsymbol{H}$ are the error and hypothesis sum of squares and products, respectively:

$$\boldsymbol{W}(x) = \boldsymbol{P}^t(x)(\boldsymbol{R}(x) - \boldsymbol{G}\boldsymbol{B}(x))^t(\boldsymbol{R}(x) - \boldsymbol{G}\boldsymbol{B}(x))\boldsymbol{P}(x) \tag{4}$$

$$\boldsymbol{H}(x) = \boldsymbol{P}^t(x)\boldsymbol{B}^t(x)\boldsymbol{C}^t(\boldsymbol{C}(\boldsymbol{G}^t\boldsymbol{G})^{-1}\boldsymbol{C}^t)^{-1}\boldsymbol{C}\boldsymbol{B}(x)\boldsymbol{P}(x)$$

$$\boldsymbol{B}(x) = (\boldsymbol{G}^t\boldsymbol{G})^{-1}\boldsymbol{G}^t\boldsymbol{R}(x)$$

$$\boldsymbol{R}(x) \equiv [\boldsymbol{r}_1^t(x)\boldsymbol{r}_2^t(x)\ldots\boldsymbol{r}_N^t(x)]^t$$

$$\boldsymbol{G} \equiv [\boldsymbol{g}_1^t\,\boldsymbol{g}_2^t\ldots\boldsymbol{g}_N^t]^t$$

Eq 3 results in a statistical parametric map $F(x)$ with values that follow, under the null hypothesis, a standard $F$ distribution with $kc$ and $kb$ degrees of freedom. This allows us to compute associated p- values characterizing the likelihood of our observations under the null hypothesis for every individual searchlight voxel.

The between-voxels contrast matrix $\boldsymbol{P}(x)$ in Eq 2 serves to focus the analyses on a particular subspace of interest characterizing specific features of the functional connectivity maps $\boldsymbol{r}_n(x)$. The choice of $\boldsymbol{P}(x)$ affords great flexibility in the specific form of fc-MVA analyses that can possibly be implemented. For example, in the simplest possible scenario, we may choose $\boldsymbol{P}(x)$ to be a constant one-dimensional projector, such as a unit-norm vector with positive weights over a single voxel or a small area, which would allow us to focus only on the connectivity with one a priori voxel or region of interest. Interestingly, in this scenario fc-MVA reduces exactly to a standard seed-based connectivity (SBC) analysis, producing statistical parametric maps $F(x)$ that characterize the connectivity between the chosen voxel or area and the rest of the brain. In contrast, in perhaps the most general scenario, we may instead choose $\boldsymbol{P}(x)$ to also be constant but now equal to the identity matrix, jointly and equally considering all target voxels. This allows us to simultaneously estimate and evaluate *any/all* aspects of the functional connectivity maps $\boldsymbol{r}_n(x)$.

Between these two extrema, there are many reasonable alternatives. For example, a spatial basis $\boldsymbol{P}(x)$ that would focus on low spatial-frequency components of connectivity profiles (e.g. [49]), one that would focus only on local connectivity with neighboring areas (e.g. a multivariate Local Connectivity measure), or one that would focus only on connectivity with all voxels within a fixed area (masked fc-MVA, e.g. connectivity with the cerebellum or any other large/

heterogeneous area). In the next section we will discuss one particular form of fc-MVA analyses that is based on a data-driven choice of spatial basis $P(x)$ focusing on rich low-dimensional representations of arbitrary functional connectivity patterns.

## Definition of functional-connectivity multivariate pattern analyses (fc-MVPA)

Functional connectivity multivariate *pattern* analyses (fc-MVPA) can be considered a particular case of functional connectivity multivariate analyses (fc-MVA), where the choice of spatial basis $P(x)$ attempts to achieve a balance between retaining high sensitivity to unknown or arbitrary effects while maintaining a good level of specificity to those features more representative of the data at hand. In particular, representative features in fc-MVPA are chosen to have maximal intersubject variability and minimal overlap (i.e. orthogonal features). This is achieved by first constructing the matrix $R(x)$ by concatenating all of the maps $r_n(x)$ for a given seed-voxel $x$ across all subjects, and then defining $P(x)$ implicitly as the right- orthogonal basis from a Singular Value Decomposition (SVD) factorization of the connectivity matrix $R(x)$:

$$S(x) \cdot D(x) \cdot P^t(x) = R(x) \qquad (5)$$

where $S(x)$ and $P(x)$ are orthogonal matrices of left- and right- singular vectors of $R(x)$, respectively, and $D(x)$ is a diagonal matrix containing the positive singular values of $R$ sorted in decreasing order. The total number of singular vectors and values in Eq 5 is equal to the number of subjects $N$, but typically this dimensionality can be further reduced to a lower value $k \leq N$ to only include the first few singular values and vectors that achieve a predefined predictive or descriptive target (e.g. those dimensions capturing on average 50% or more of the total covariance in the patterns of functional connectivity across subjects).

Conceptually, this particular choice of basis in fc-MVPA has a very important benefit, as the resulting *eigenpatterns*, defined as the columns of the resulting matrix $P(x)$, have a meaningful interpretation as those patterns that best characterize the observed heterogeneity across subjects in functional connectivity with an individual seed voxel. In particular the squared eigenvalues

$$\xi(x) \equiv diag(D^2(x))/trace(R(x) \cdot R^t(x)) \qquad (6)$$

represent the portion of the total intersubject covariance $R(x) \cdot R^t(x)$ in connectivity maps that lies within the dimensions characterized by each individual eigenpattern, and by the nature of SVD these values are maximal (i.e. there is no other $k$-dimensional subspace containing a larger percentage of the total covariance of the data than the subspace spanned by the first $k$ eigenpatterns, for any value $k$).

In this context, the values $s_n(x)$, which we will refer to in this manuscript as *eigenpattern scores*, and which are defined as the rows of the left-singular matrix $S(x)$, define an optimal linear low-dimensional representation of the original data $r_n(x)$ for each subject, meaning that we can always linearly reconstruct the high-dimensional data $r_n(x)$ from its low-dimensional representation $s_n(x)$ with minimal error.

Mathematically, this approach is similar to functional PCA [50] or to the group-level dimensionality reduction step in ICA [13] which helps reduce noise, simplify the analyses, and increase the interpretability of the resulting ICA components, but the main difference is that in fc-MVPA dimensionality reduction is performed separately for each individual seed-voxel $x$. Because the dimensionality reduction step in fc-MVPA is only tasked with characterizing the heterogeneity in functional connectivity patterns between one individual voxel $x$ and the rest of the brain, while in PCA/ICA the dimensionality reduction step is tasked with

simultaneously characterizing the heterogeneity in functional connectivity patterns between *all* pairs of voxels, the former can achieve a considerably more compact representation, where fewer components will explain a larger portion of that heterogeneity (as will be illustrated in the next section). In addition, the ability to obtain such low-dimensional characterization in a way that is specific to each anatomical location offers a considerably richer representation of intersubject heterogeneity compared to other similar but global approaches, such as PCA or ICA.

In the context of brain-wide connectome analyses, using a simple change of basis allows us to simplify the fc-MVA multivariate general linear model and null hypothesis in Eq 2. In particular, by right-multiplying Eq 2 by the matrix $P(x) \cdot D^{-1}(x)$ we develop an equivalent lower-dimensional fc-MVPA general linear model and hypothesis of the form:

functional connectivity MultiVariate Pattern Analyses (fc-MVPA)

$$\forall x \qquad s_n(x) = g_n \cdot \tilde{B}(x) + \varepsilon_n(x) \cdot \tilde{\Sigma}(x)$$
$$\text{Null hypothesis}: \qquad C \cdot \tilde{B}(x) = 0 \tag{7}$$

This is exactly the same model as in Eq 2 but expressed only within a lower-dimensional subspace represented by the eigenpattern scores $s_n(x)$, instead of the original higher-dimensional connectivity maps $r_n(x)$. In this context, the eigenpattern scores $s_n(x)$ represent what has also been referred to as Multivariate Connectivity maps (MCOR) [51], a voxel-specific low-dimensional multivariate representation of the pattern of functional connectivity between a voxel and the rest of the brain. Similarly, $\tilde{B}(x)$ and $\tilde{\Sigma}(x)$ in Eq 7 are also equal to their Eq 2 counterparts simply projected onto the subspace defined by $P(x)$. The reduced dimensionality allows us to simplify the computational implementation of these analyses considerably. For example, the eigenpattern scores $s_n(x)$ can now be simply stored as multiple whole-brain volumes, with one volume or image per component and per subject, and shared across multiple second-level analyses. This is in contrast with the considerably larger vectors $r_n(x)$ which cannot be easily stored (e.g. one whole-brain volume per subject *and per target voxel y*). In addition, the eigenpattern scores $s_n(x)$ are defined independently of the predictor vectors $g_n$, so they not only offer a model-free characterization of the heterogeneity in the data, but the same eigenpattern scores can also be used in multiple different group-level analyses of the same data. Last, the reduced dimensionality of Eq 7 also allows the covariance $\tilde{\Sigma}^2(x)$ across eigenpattern scores to be fully estimable from a limited number of samples, whereas the original covariance $\Sigma(x)$ across voxels in Eq 2 could very rarely be so estimated with full rank. Because of this, the effect of the within-subjects covariance in the resulting null hypothesis *F* statistics at each individual searchlight voxel does not need to be approximated (e.g. using Satterthwaite approximation as in Eq 3), allowing us to rely instead on a more sensitive Likelihood Ratio statistic (LRT) of the form [52]:

$$F(x) = \frac{d}{ac} \cdot \frac{1 - \lambda^{1/e}}{\lambda^{1/e}} \sim F_{ac,d} \tag{8}$$

$$\lambda = \frac{|W|}{|W + H|}$$

$$a \equiv rank(P(x))$$

$$b \equiv N - rank(\mathbf{G})$$

$$c \equiv rank(\mathbf{GC}^t)$$

$$d \equiv \left( b - \frac{a - c + 1}{2} \right) e - \frac{ac}{2} + 1$$

$$e \equiv \sqrt{\frac{a^2 c^2 - 4}{a^2 + c^2 - 5}}$$

where $\lambda$ is Wilks' Lambda statistic, $a$ is typically equal to $k$, the number of selected eigenpatterns ($1 < a < b$), $b$ is the error degrees of freedom, $c$ is the hypothesis degrees of freedom, and $\mathbf{W}$ and $\mathbf{H}$ are the same error and hypothesis sum of square matrices as in Eq 4 computed over the subspace $\mathbf{P}(x)$. As before, the resulting $F(x)$ values follow, under the null hypothesis, a standard $F$ distribution with $ac$ and $b$ degrees of freedom, for each individual searchlight voxel $x$.

To summarize, Fig 1 illustrates the idealized fc-MVPA procedure. For every searchlight voxel $x$ we first compute the functional connectivity maps $\mathbf{r}_n(x)$ between this voxel and the rest of the brain for every individual subject (Fig 1 top left), and use Eq 5 to compute a reduced set of eigenpattern scores $\mathbf{s}_n(\mathbf{x})$ best characterizing relevant spatial features of these maps across subjects (represented in Fig 1 top right). Once each subject's functional connectivity profiles are represented in terms of their lower-dimensional associated eigenpattern scores $\mathbf{s}_n(\mathbf{x})$, group-level functional connectivity analyses proceed normally by entering these scores into a standard General Linear Model (Eq 7). This model evaluates at this searchlight location $x$ any hypothesis of the form $\mathbf{C} \cdot \tilde{\mathbf{B}}(x) = \mathbf{0}$ using LRT (Eq 8), allowing us to make inferences about the shape of the functional connectivity maps that these scores represent. Last, this procedure is then simply repeated for each searchlight voxel $x$, sequentially constructing a statistical parametric map $F(x)$ characterizing the results of this inferential procedure across the entire brain.

The general fc-MVPA procedure may be seen as computationally prohibitive, particularly for whole-brain analyses using relatively small voxel sizes (e.g. isotropic 2mm voxels), since the computational load scales quadratically with the total number of voxels under consideration, it appears to require the computation of entire voxel-to-voxel connectivity matrices, and it effectively performs close to 200,000 whole-brain PCA analyses (one per seed voxel) characterizing the intersubject heterogeneity of seed-based connectivity maps. Despite this, there are several mathematical tricks that can be used to reduce the complexity of the necessary computations by several orders of magnitude. In particular, in S1 Appendix we describe how to more efficiently compute the eigenpattern scores $s_n(x)$ directly from the original BOLD timeseries in a way that will instead scale only linearly with the number of voxels, and without the need at any point to compute or store entire voxel-to-voxel connectivity matrices. In the analysis examples below we use this approach to efficiently compute fc-MVPA analyses on hundreds of subjects with minimal computation effort.

## Results and discussion

### Fc-MVPA brain-wide connectome inferences: interpretation and examples

Group-level analyses of the eigenpattern scores $\mathbf{s}_n(\mathbf{x})$ enable statistical inferences at the level of individual searchlight voxels evaluating the form or shape of the connectivity patterns with

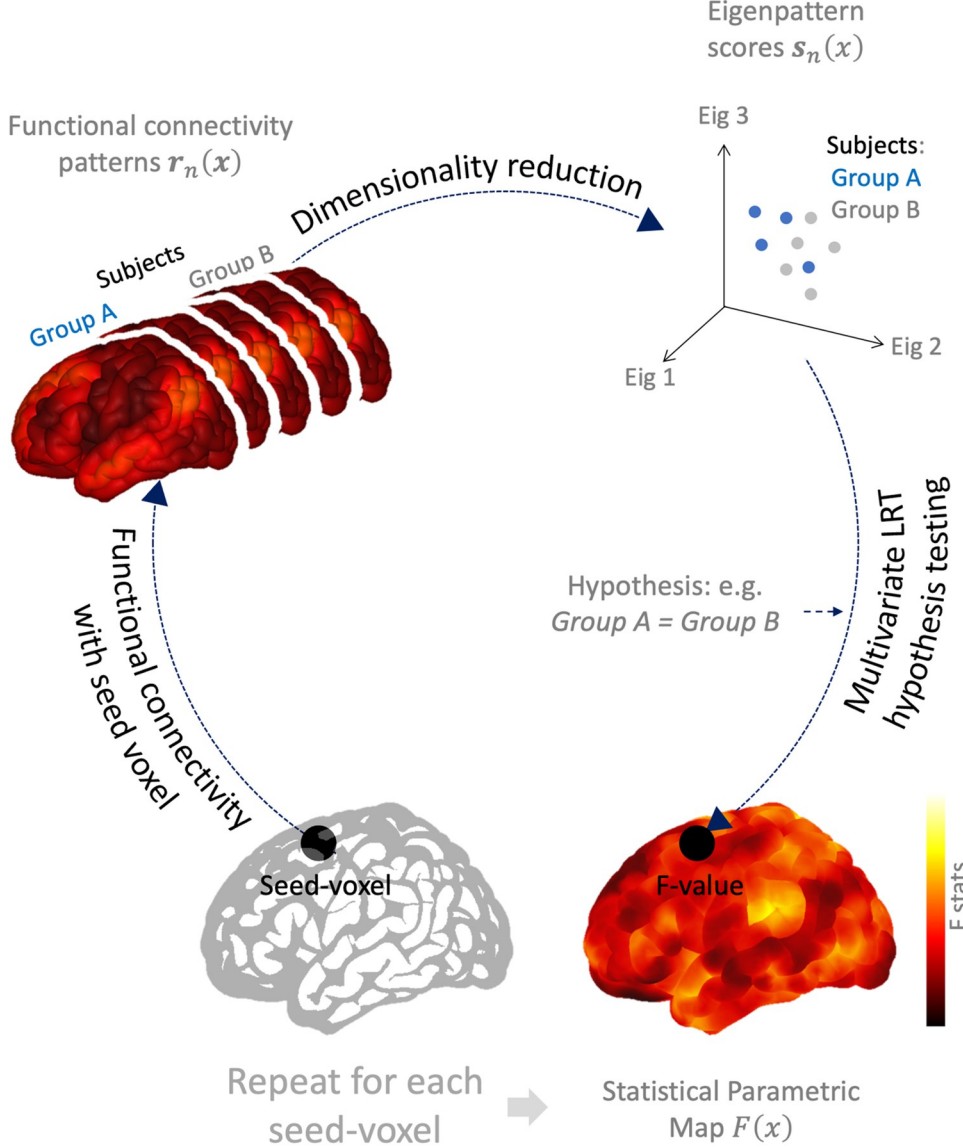

**Fig 1. Schematic representation of functional connectivity multivariate pattern analyses (fc-MVPA).** For each voxel, fc-MVPA analyses compute the functional connectivity maps between this seed/source voxel and the entire brain (Top-left; $r_n(x)$ in Eq 2) separately for each individual subject. Each subject functional connectivity map is then characterized by a lower dimensional eigenpattern scores (dots in top-right graph; $s_n(x)$ in Eq 5). This representation is chosen in a way that captures as well as possible the observed voxel-specific variability in functional connectivity maps across subjects. A multivariate test is then performed on the resulting lower-dimensional eigenpattern scores to ascertain potential between- or within- subjects effects of interest (e.g. differences between subjects or between conditions in functional connectivity at the original seed/source voxel). This process is then repeated for every source voxel to identify regions that show brain-wide between- or within- subjects differences in functional connectivity.

each voxel. In particular, for any individual hypothesis (e.g. group A = group B) the fc-MVPA procedure will produce a statistical parametric map $F(x)$ evaluating that hypothesis separately at each individual searchlight voxel $x$. In order to test brain-wide connectome hypotheses it is still necessary to control the resulting maps $F(x)$ for multiple comparisons across the total number of tests evaluated (one test per voxel). Fortunately, this can be done using the same nonparametric cluster-level inferential procedures that are common in standard analyses of

functional activation, such as cluster-mass or TFCE statistics based on permutation / randomization analyses [5,6,53]. These approaches allow us to compute statistics as well as associated familywise error corrected p-values for individual **clusters** of contiguous searchlight voxels in the statistical parametric map $F(x)$, supporting cluster-level inferences with meaningful false positive control (e.g. controlling the likelihood of observing or more false positive clusters across the entire brain below 5%, for a familywise control procedure, or controlling the rate of false positive clusters below 5% among all significant clusters, for a False Discovery Rate control procedure).

Another important choice that remains when using fc-MVPA in the context of brain-wide connectome inferences is to select $k$, the dimensionality of $s_n(x)$ or the number of eigenpattern scores used to represent functional connectivity at each voxel. As it will be discussed in more detail in the *Simulations* section below, there is no "correct" choice of this parameter, and fc-MVPA inferences remain valid for all possible values of this parameter. Nevertheless it is important that this choice is made a priori and justified (e.g. from prior literature), and, if different values are tested/evaluated, the results of all these different evaluations should be reported (rather than reporting only the value that produces the best results for a particular analysis, which would inflate the chance of false positives). Regarding the sensitivity of fc-MVPA inferences, choosing a low value of $k$ can be expected to improve sensitivity to detect relatively large or widespread effects of interest such as inter-network connectivity differences, while choosing a higher value of $k$ can improve our ability to detect relatively smaller or marginal effects such as connectivity with smaller areas or subnetworks. In the absence of assumptions about the extent of the expected effects, a reasonable balance is to scale the choice of $k$ with the dataset size (e.g. suggested 5:1, 10:1, or 20:1 ratio between N:k, the number of subjects in the analysis and the number of eigenpattern scores retained [54–56]), in order to maintain a reasonable sensitivity to identify large effects in small samples, and comparatively finer details in the analysis of larger samples. As with any other preprocessing or analysis choices, as long as the choice of $k$ is made a priori, statistical inferences will remain valid. If, on the other hand, the value of $k$ is selected a posteriori as the one that produced the "best" results among several possible values evaluated, an appropriate multiple-comparison correction should be used for statistical inferences (e.g. using a Bonferroni corrected cluster-level threshold p-FWE<0.05/10 if the results were selected among 10 different choices of $k$ values, or using split or cross-validation procedures such as choosing $k$ as the value that produces the "best" results in one half of the subjects and then basing statistical inferences on the analysis of the other half using the selected $k$ value). For new analyses, and in the absence of any other rationale (e.g. based on prior literature, N:k ratios, or expected extent of effects) we recommend using a value of $k = 10$, as that seems to suffice to cover a large proportion of the intersubject variability in functional connectivity profiles (e.g. see fc-MVPA eigenpatterns section below). In all cases we encourage researchers to evaluate and report the robustness of their results to different choices of this parameter (e.g. as exploratory post-hoc analyses, without a need for additional multiple comparison corrections), as that will help other researchers build upon those results in future analyses and the field converge toward useful conventions.

Last, regarding the interpretation of fc-MVPA results, when reporting statistical inferences from fc-MVPA brain-wide connectome analyses, those inferences should be if possible framed regarding the **patterns** of connectivity between each voxel or cluster and the rest of the brain. For example, when using fc-MVPA to evaluate the difference in connectivity between two groups of subjects, if the fc-MVPA procedure above produces one supra-threshold cluster with corrected significance level below p < .05 that should be interpreted as indicating that the pattern of connectivity between this cluster and the rest of the brain is (significantly) different between the two groups. The fc-MVPA method does not afford further spatial specificity in

the resulting statistical inferences, but it is still useful to report measures of effect-size characterizing the patterns of connectivity within each individual significant cluster, as a way to suggest possible interpretations and future analyses.

Effect-sizes in GLM analyses are typically represented by linear combinations of the estimated regressor coefficients $\boldsymbol{B}$, and, specifically in the context of hypothesis testing, linear combinations of the form $\boldsymbol{c} \cdot \boldsymbol{B}$, as these measures quantify the extent of the observed departures from the null hypothesis ($\boldsymbol{c} \cdot \boldsymbol{B} = \boldsymbol{0}$). The interpretation of effect-sizes in GLM depends naturally on the choice of hypothesis being evaluated. For example, for a GLM two-sample t-test comparing connectivity between two groups, effect-sizes of the form $\boldsymbol{c} \cdot \boldsymbol{B}$ would represent the difference in connectivity (e.g. average differences in r- values) between the two groups, while for a GLM regression analysis evaluating the association between some behavioral measure and connectivity strength, effect-sizes of the form $\boldsymbol{c} \cdot \boldsymbol{B}$ would represent the slope of the regression line approximating the observed associations. Note that in both cases an effect-size of zero would represent the null hypothesis (of no differences or no associations, respectively). In the analysis of the statistical parametric map $F(x)$, for any significant cluster $\Omega$ (group of contiguous suprathreshold voxels with a cluster-level corrected p-value below the chosen family-wise error threshold), we recommend reporting the effect-sizes $\boldsymbol{h} \equiv \boldsymbol{c}_j \cdot \tilde{\boldsymbol{B}}$ separately for each meaningful between-subjects contrast $\boldsymbol{c}_j$ (e.g. individual rows of the contrast matrix $\boldsymbol{C}$). Effect sizes can be reported as a vector of eigenpattern weights ($\boldsymbol{h}_{\text{eig}}(\Omega)$), or as a whole-brain projected map ($\boldsymbol{h}_{\text{map}}(\Omega)$):

fc-MVPA effect-sizes at location $\Omega$

$$\boldsymbol{h}_{\text{eig}}(\Omega) \equiv \sum\nolimits_{x \in \Omega} \boldsymbol{c}_j \cdot \tilde{\boldsymbol{B}}(x) \tag{9}$$

$$\boldsymbol{h}_{\text{scores}}(\Omega) \equiv \sum_{x \in \Omega} \boldsymbol{c}_j \cdot \tilde{\boldsymbol{B}}(x) \cdot \boldsymbol{S}^t(x) = \sum_{x \in \Omega} \boldsymbol{h}_{eig}(x) \cdot \boldsymbol{S}^t(x)$$

$$\boldsymbol{h}_{\text{map}}(\Omega) \equiv \sum_{x \in \Omega} \boldsymbol{c}_j \cdot \tilde{\boldsymbol{B}}(x) \cdot \boldsymbol{D}(x) \cdot \boldsymbol{P}^t(x) = \sum_{x \in \Omega} \boldsymbol{h}_{scores}(x) \cdot \boldsymbol{R}(x)$$

The effect-size measure $\boldsymbol{h}_{\text{eig}}(\Omega)$ is a vector, with one element per eigenpattern, estimated separately at each location $\Omega$. It represents the effect-size of a group-level analysis contrast of interest $\boldsymbol{c}_j \cdot \tilde{\boldsymbol{B}}(x)$ evaluated separately for each individual eigenpattern (columns of $\tilde{\boldsymbol{B}}$). For example, if the group-level analysis was a two-sample t-test comparing connectivity between two subject groups, then the k-th element in the $\boldsymbol{h}_{\text{eig}}(\Omega)$ effect-size vector will evaluate what is the difference in the k-th eigenpattern scores at location $\Omega$ between these two groups.

Similarly, and perhaps more directly interpretable, the effect-size measure $\boldsymbol{h}_{\text{map}}(\Omega)$ represents the same contrast but is now evaluated separately at each individual voxel. In the example above, the value of $\boldsymbol{h}_{\text{map}}(\Omega)$ at a particular voxel will represent the difference in functional connectivity between $\Omega$ and this voxel between the two subject groups analyzed. It should be noted that a very similar whole-brain projected map of effect-sizes $\boldsymbol{h}_{map}(\Omega)$ can also be computed from the voxel-level effect-sizes of a post hoc analysis that would evaluate the same group-level model as in Eq 7 but this time focusing on the seed-based connectivity maps (SBC) associated with seeds defined from each individual significant cluster $\Omega$. As in any post-hoc analysis, p-values derived from these SBC post-hoc analyses will be partially inflated due to selection bias and should not to be used to make secondary inferences regarding individual connections within the reported patterns. Despite this limitation, post-hoc SBC analyses on

the same dataset offer a simple and perfectly valid alternative approach for reporting fc-MVPA effect sizes within each significant cluster, while, when performed on an independent dataset, they also offer a natural method to further probe specific aspects of these connectivity patterns. For those interested, the resulting $h_{map}(\Omega)$ effect-sizes following this approach can be shown to be equal to those derived from Eq 9 in the limit when the number of eigenpatterns retained equals the total number of subjects in the study, simplifying the variable $h_{scores}(\Omega)$ to a constant vector independent of the location $\Omega$:

$$h_{scores}^{\infty}(\Omega) = c_j \cdot (G^t \cdot G)^{-1} \cdot G^t \qquad (10)$$

Effect-size measures such as those described in Eqs 9 and 10 represent post-hoc estimates, and as such they should always be understood to contain a certain amount of bias. Their application is mainly for interpretation purposes and for hypothesis building. In other contexts, when the accuracy of these estimates may be essential, a cross-validation approach may be used where, for example, the clusters $\Omega$ may be computed from an initial General Linear Model (Eqs 7 and 8) that includes only data from a subset of subjects, while the effect-size estimates (Eqs 9 and 10) may be computed using a second GLM that includes data from a separate/independent subset of subjects.

As an illustration of fc-MVPA brain-wide connectome inferences, we analyzed gender differences in resting state functional connectivity using the Cambridge 1000-connectomes dataset (n = 198, see S2 Appendix for a description of this dataset preprocessing and fc-MVPA analysis methods [57–66]). The question that these analyses ask is whether there are *any* differences across the entire voxel-to-voxel functional connectome between male and female subjects. To answer this question, we performed fc-MVPA analyses focusing on the first 10 eigenpatterns (an approximate 20:1 subjects to eigenpattern ratio), entering the corresponding eigenpattern scores into a second-level group analysis evaluating a multivariate ANCOVA test with gender as a between-subjects factor and subject motion (average framewise displacement) as a control variable. The resulting statistical parametric maps were thresholded using Threshold Free Cluster Enhancement [6] (TFCE, with default H = 1, E = 0.5 values) at a familywise error corrected 5% false positive level.

The results, shown in Fig 2 show a large number of areas with significant gender-related differences in connectivity (p-FWE < .05, shown as yellow and black areas in the center image). Given the abundance of areas showing significant gender effects, for illustration purposes we focused our description only on a subset of cortical regions showing some of the strongest effects (TFCE>200; p-FWE < .001, highlighted in black in Fig 2 center image). For each cluster in this reduced subset, we computed effect-size maps $h_{map}(\Omega)$ characterizing the pattern of gender-related differences in connectivity with each cluster (displayed in Fig 2 as a circular array of brain displays), with yellow indicating higher connectivity with this cluster in male compared to female subjects, and blue indicating higher connectivity in female compared to male subjects.

Some of the strongest effects were visible in the bilateral Occipital Pole visual areas. A left hemisphere cluster centered at MNI coordinates (-22,-94,+4) mm showed a pattern of increased connectivity with Default Mode Network (DMN) and increased anticorrelations with Salience Network (SN) areas in male subjects (see Fig 2 Occipital Pole plot). A similar pattern (not shown) was present in another cluster in right hemisphere Occipital Pole areas (+28,-82,+2). Similarly, there were significant gender effects in several DMN areas, such as Medial Prefrontal Cortex (+6,+54,-12) and Precuneus (+18,-72,+32), showing a similar pattern of stronger connectivity with visual and sensorimotor areas (shown in yellow in Fig 2 Medial Frontal Cortex and Precuneus plots) in male subjects compared to stronger connectivity

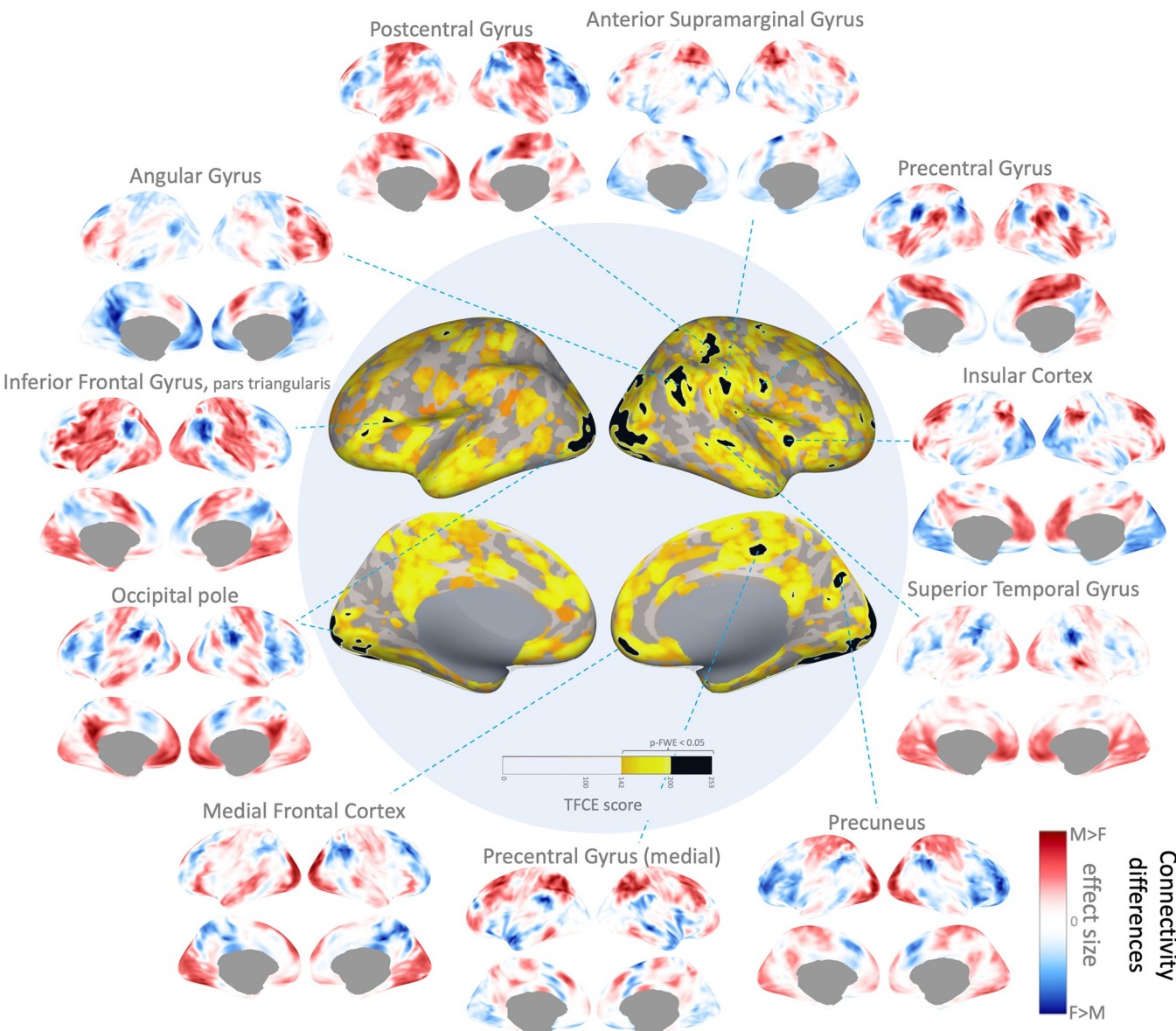

**Fig 2. Fc-MVPA results evaluating gender-related differences in connectivity.** Central figure shows left- and right- hemisphere medial (bottom) and lateral (top) views of the main fc-MVPA results showing areas with significant gender-related differences in functional connectivity (highlighted in yellow and black, TFCE statistics p-FWE<0.05). Among all significant results a reduced subset showing some of the strongest effects are highlighted in black, and the effect-sizes within these areas (pattern of differences in connectivity with each area between male and female subjects) are shown in the additional circular plots (yellow indicating higher connectivity in male compared to female subjects, and blue indicating higher connectivity in female compared to male subjects).

(weaker anticorrelations) with SN and attention areas in female subjects (shown in blue in same plots).

In the left hemisphere, Inferior Frontal Gyrus pars triangularis (-54,+22,+14) showed a pattern of stronger connectivity in female subjects with frontoparietal network areas and with Inferior Temporal Cortex (shown in blue in Fig 2 Inferior Frontal Gyrus plot).

In the right hemisphere, there was a cluster of regions in the Temporal Parietal Occipital Junction that also showed strong gender-related differences in connectivity. Lateral superior Postcentral Gyrus (+36,-32,+48) showed a mixed pattern of increased integration with other

Dorsal Attention areas in female subjects, compared to increased connectivity with Central Sulcus, including somatosensory and motor areas in male subjects. A cluster in superior Angular Gyrus centered at coordinates (+50,-54,+42) mm showed increased integration with medial prefrontal and posterior cingulate areas in female subjects, and increased integration with lateral prefrontal and reduced anticorrelations with insular areas in male subjects. Relatedly, a cluster in the mid Insular Cortex (+38,+10,+2) showed a similar pattern of higher connectivity with angular Gyrus and other DMN areas in male subjects. Another relatively proximal cluster in the Anterior Supramarginal Gyrus (+60,-26,+30) also showed increased local associations with superior postcentral areas in male subjects. Posterior Superior Temporal Gyrus (+48,-26,-2) showed a pattern of higher connectivity (mixed with reduced anticorrelations) with fronto-parietal areas in female subjects (shown in blue in Fig 2 Superior Temporal Gyrus plot), compared to a similar pattern of stronger local connectivity in male subjects (shown in yellow in same plot).

Last, Medial Precentral Gyrus areas (+16,-26,+40) showed relatively higher integration with SN or ventral attention network in female subjects. In contrast, lateral Precentral Gyrus areas (+54,-4,+22) showed higher integration with the same networks in male subjects, while in female subjects this area showed stronger local correlations (shown in blue in Fig 2 Precentral Gyrus plot).

From a validation and generalization perspective, it is interesting to question whether the same or similar results would have been observed if, instead of using 10 eigenpattern scores, based on a conservative suggestion to maintain an approximately 20:1 ratio between subjects and eigenpatterns, we would have chosen a different number. To that end we repeated the previous group-level analyses evaluating gender differences in connectivity but now using different number of eigenpattern scores, ranging from 1 to 100, and compared the resulting fc-MVPA statistic parametric maps $F(x)$.

The results (Fig 3 top) show very similar $F(x)$ statistics when varying the number of eigenpatterns around the k = 10 value selected for our original analyses. In addition, the distribution of resulting statistics across the entire brain (Fig 3 bottom) shows high sensitivity across the entire range of evaluated k values, consistent with the sensitivity simulations in the sections below, and with average sensitivity peaking at k = 50 (approximately a 4:1 ratio in subjects to eigenpatterns) for detecting gender effects in this dataset. While there were several areas like superior Postcentral Gyrus where the statistics peaked at relatively low number of eigenpatterns, suggesting that the effects in these areas may be best represented by the first few fc-MVPA eigenpatterns (i.e. they may be better described in terms of common/large sources of variability across subjects), there were also many areas where the $F(x)$ statistics peaked when using a large number of eigenpatterns (e.g. 50 or above), suggesting that there may still be widespread gender differences in functional connectivity beyond those highlighted in our original analyses and described in Fig 2 that are better expressed in some of the higher-order fc-MVPA eigenpatterns (i.e. representing more subtle or less common sources of intersubject variability).

## Fc-MVPA eigenpatterns $P(x)$: interpretation and examples

In addition to enabling brain-wide connectivity inferences, fc-MVPA estimates a model-free representation of the observed intersubject variability in functional connectivity in terms of the resulting eigenpatterns $P(x)$, which can be useful on its own. In this context, the *eigenpatterns*, defined as the columns of the voxel-specific matrix $P(x)$, represent a set of mutually orthogonal spatial patterns, different for each voxel, that best characterize the diversity across subjects in functional connectivity between this voxel and the rest of the brain. By convention

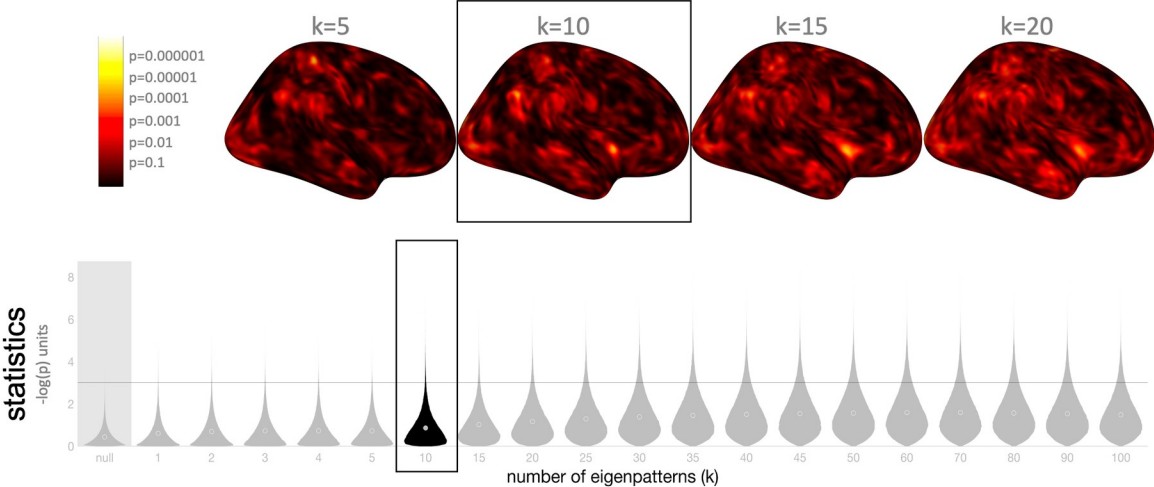

**Fig 3. Selecting different number of fc-MVPA eigenpatterns.** Difference in fc-MVPA statistic parametric maps evaluating gender differences in connectivity, when varying $k$, the number of fc-MVPA eigenpatterns used in the analysis, from k = 1 (left) to k = 100 (right). For reference, the original results shown in Fig 2 used k = 10 (highlighted here inside black box). **Top**: Statistic parametric maps with color coding showing voxel-level $-\log_{10}(p)$ values for four different choices of k (from 5 to 20). The results show consistent statistic parametric maps across different $k$ values. **Bottom**: Distribution of fc-MVPA statistics across all gray matter voxels with $k$ ranging from 1 to 100, compared to null hypothesis distribution (shown in leftmost 'null' histogram). The results indicate high sensitivity across the entire range of evaluated $k$ values, with sensitivity peaking at around k = 50 (close to a 4:1 ratio in subjects to eigenpatterns) for detecting widespread gender effects in this dataset.

they are sorted in descending order based on the proportion of the total intersubject covariance explained by each eigenpattern.

In practice, while it is straightforward to compute and store the eigenpattern scores $s_n(x)$, storing the entire set of eigenpatterns $P(x)$ can be particularly demanding as it consists of a set of orthogonal whole-brain maps for each individual voxel. Luckily it is simple to define $P(x)$ analytically at any individual voxel $x$ from its associated eigenpattern scores as:

$$P(x) \propto \sum_n r_n^t(x) \cdot s_n(x) \tag{11}$$

which can be generalized to define characteristic eigenpatterns over small homogeneous areas by integrating the corresponding voxel-specific eigenpatterns:

fc-MVPA eigenpatterns at location $\Omega$

$$P(\Omega) \propto \sum_n \sum_{x \in \Omega} r_n^t(x) \cdot s_n(x) \tag{12}$$

This can also be useful in the context of a cross-validation framework where it may be necessary to compute eigenpattern scores $s_n(x)$ for a new, yet unseen, set of subjects. This can be done using Eq 12 to first characterize $P(x)$ from the original data, and then Eq 5b to compute $s_n(x)$ from the new subjects' connectivity maps.

In general, reporting and describing the fc-MVPA eigenpatterns in Eq 12 over a small area, along with the proportion of the total covariance explained by each eigenpattern at this area, allows to gain a better understanding of the main factors affecting intersubject heterogeneity in functional connectivity between this area and the rest of the brain.

For example, from the analysis of the same resting state data of 198 subjects in the Cambridge dataset, the map $\xi_1(x)$ shown in Fig 4 (bottom) describes the proportion of the total

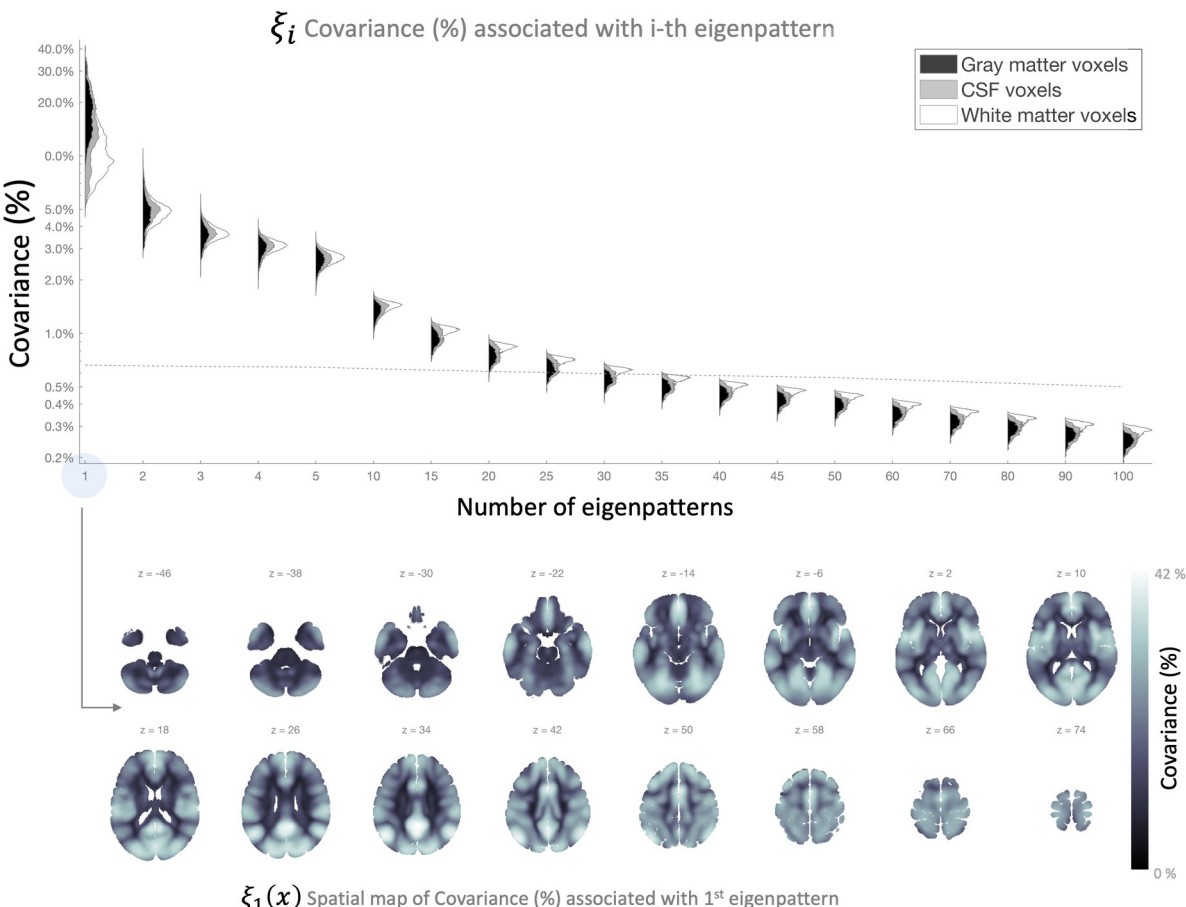

**Fig 4. Percentage of total covariance associated with each fc-MVPA eigenpattern. Top**: histogram of $\xi_k(x)|1{\leq}i{\leq}100$ values, percentage of the total covariance explained by each of the first 100 eigenpatterns. Histograms are further broken down by the most likely tissue class (gray matter in black, CSF areas in grey, and white matter in white) at each individual voxel as defined by SPM's tissue probability map templates. **Bottom**: spatial map $\xi_1(x)$ showing the proportion of the total intersubject covariance explained by each voxel's first eigenpattern (a measure of the overall intersubject homogeneity in functional connectivity patterns at each voxel; see text for details).

intersubject covariance explained by each voxel's first eigenpattern. In this sample, $\xi_1(x)$ values range between 5% and 42% across different voxels. The values $\xi_1(x)$ associated with the first eigenpattern are particularly interesting because they provide a simple measure of the overall intersubject homogeneity of the connectivity maps at each voxel (with higher values indicating higher homogeneity), as the first eigenpattern can often be expected to lie in the direction of the average pattern of connectivity with that voxel (as illustrated in the analyses below). Overall, this sample $\xi_1(x)$ map shows high contrast between gray matter areas and other tissue classes, with higher values within gray matter areas, particularly those located along cortical gyra, and relatively lower values for areas located deeper into cortical sulci. Some of the regions that show the highest homogeneity include Medial Prefrontal, Posterior Cingulate, or Lateral Parietal areas, part of the Default Mode Network (DMN), as well as anterior Insula and other Salience Network (SN) areas. In contrast, cerebellar, subcortical, and Limbic Network areas are some of the regions that show the most heterogeneous functional connectivity profiles across subjects (lowest $\xi_1(x)$ values).

Looking at the contribution of additional eigenpatterns beyond the first one, Fig 4 (top) shows histograms of the values $\xi_k(x)$, the percentage of the total intersubject covariance

explained by each of the first 100 eigenpatterns. The distribution shows a strongly anisotropic covariance (for reference, the dashed line simulates the expected percent covariance values associated with each eigenpattern if the intersubject covariance was isotropic, with equal covariance along all subject dimensions and approximate 100-voxel spatial resels). In general, approximately 20 eigenpatterns are associated with higher-than-average covariance values. The first 5 eigenpatterns combined explain a range between 18% and 51%, 10 eigenpatterns between 28% and 57%, and 20 eigenpatterns between 39% and 64%, of the total intersubject covariance (from a maximum of 197 eigenpatterns that could be theoretically computed from this sample).

While $\xi_k(x)$ maps allow us to explore how the degree of anisotropy varies across different areas, it is often also of interest to display the actual eigenpatterns $P(x)$ at some particular representative locations, in order to better understand the *shape* of that intersubject covariance. Fig 5 shows the first 5 eigenpatterns at 14 example locations. These 14 locations were manually chosen to illustrate some of the similarities and differences across different locations in intersubject variations of functional connectivity patterns. They were selected among the set of all local maxima of the cumulative $\sum_{k=1}^{5} \xi_k(x)$ maps (shown in Fig 5 center image for reference) trying to cover most of the larger clusters observed there. As this figure illustrates, the first eigenpattern across different locations (shown in the leftmost portion of each individual-region display in Fig 5) often reflects a pattern mimicking the average connectivity between each location and the rest of the brain. For example, the first eigenpattern at the Posterior Cingulate gyrus, a region part of the Default Mode Network (DMN), reflects an arrangement very similar to the expected pattern of positive and negative associations with the DMN, and the same arrangement appears in the first eigenpattern at other distant but related locations, such as Frontal Medial Cortex. Similarly, the first eigenpattern at Anterior Insula or Anterior Cingulate also shows similar profiles mimicking Salience or Ventral Attention Network connectivity. In contrast, second- and higher- order eigenpatterns, even from regions that are part of the same network, show noticeable differences in their profiles, possibly indicating nonoverlapping sources of intersubject variability beyond simple within-network connectivity variations. Other regions, in contrast, show eigenpatterns that reflect perhaps competing contributions. For example, in Paracingulate Gyrus the first eigenpattern reflects sources of variability in connectivity with nearby DMN areas while the second eigenpattern possibly reflects variability in connectivity with Anterior Cingulate and Medial Prefrontal regions.

From a validation perspective, it is interesting to note that the covariance explained by the first few fc-MVPA eigenpatterns and represented by the $\xi_k(x)$ values is always, by nature of the fc-MVPA SVD procedure, larger than what could be achieved by any other alternatively-defined eigenpatterns. In particular, it is larger than the spatial patterns that would result from a standard ICA or PCA decomposition of the same functional data. In order to highlight this, we computed on this dataset a Principal Component Analysis in CONN using the same concatenative approach and dimensionality reduction steps as in Calhoun et al. [13] to produce a set of representative components sorted by decreasing explanatory power (shown in Fig 6 bottom). We then computed, for each of these components, the covariance in functional connectivity with each individual voxel along those dimensions, and plotted histograms of the resulting cumulative variance as a function of the number of components retained (Fig 6 top). As expected, the covariance explained cumulatively by the first $k$ fc-MVPA eigenpatterns at each individual voxel (shown in gray in Fig 6 top) is always equal to or larger than the covariance explained cumulatively by the same number of PCA components (shown in black in the same plots). While this is a necessary consequence of the SVD properties as used in the context of fc-MVPA, it is important to note that in particular this implies that if we would like to

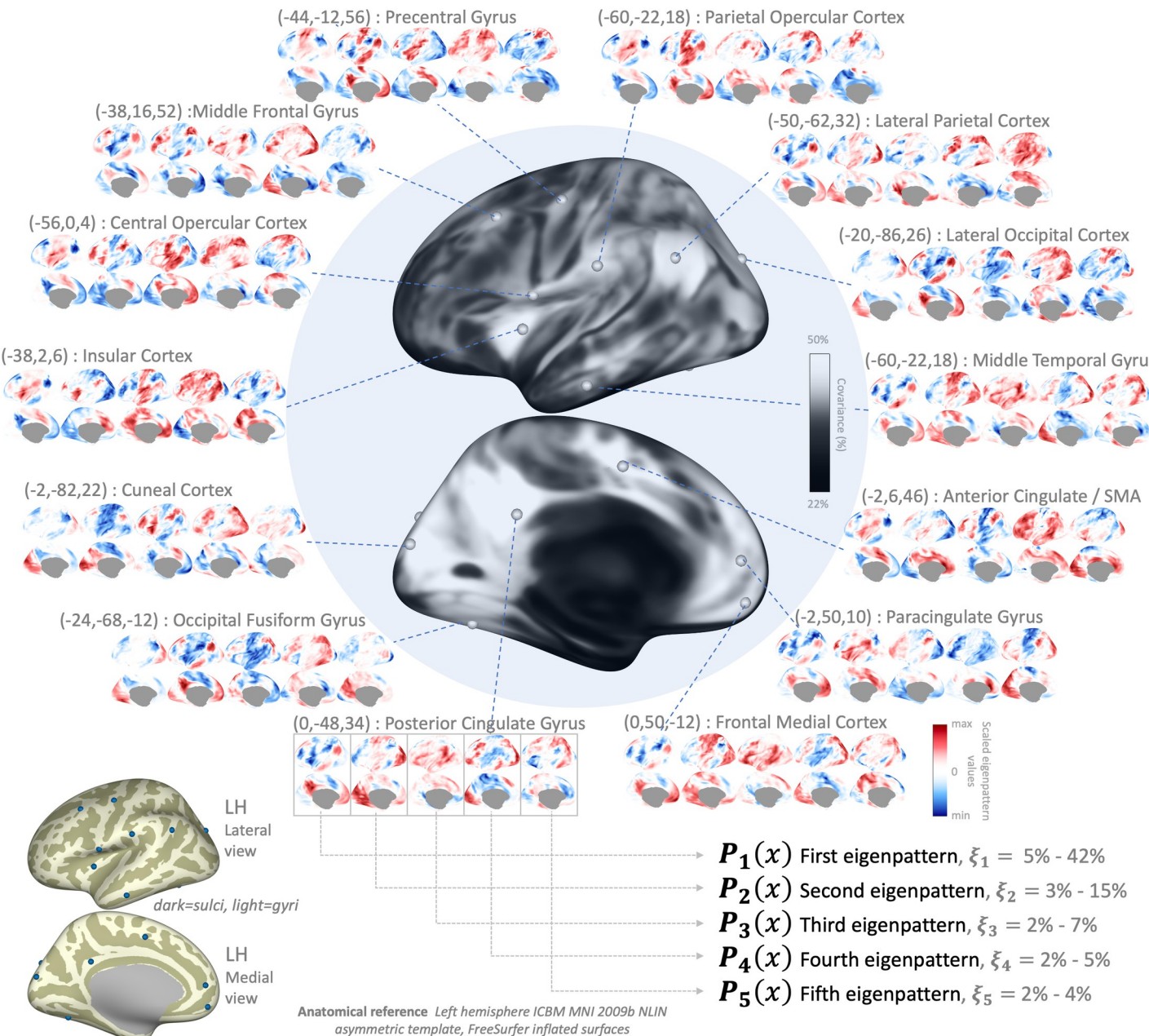

**Fig 5. First 5 fc-MVPA eigenpatterns, characterizing the principal components of the local intersubject heterogeneity in functional connectivity maps.** The central display shows the cumulative total covariance in functional connectivity patterns explained by the first 5 eigenpatterns at each voxel (colormap values range between 22%/black to 50%/white). The first five eigenpatterns at 14 manually-defined example locations are shown in a circular display. In each of these plots, eigenpatterns range from first/left to fifth/right, and each eigenpattern is shown projected to a left hemisphere lateral (top plot) and medial (bottom plot) views, on a relative color scale ranging from blue (highest negative values for each eigenpattern) to yellow (highest positive values).

characterize the functional connectivity pattern at each voxel using a reduced fixed number of scores, then the representation produced by the fc-MVPA eigenpattern scores would always be more efficient (it would better approximate the functional connectivity data) than an equally-sized multivariate representation produced by characterizing each voxel connectivity in terms of network-level properties (at least for the general class of linear transformations projecting each connectivity pattern onto multiple networks, including those resulting from Principal

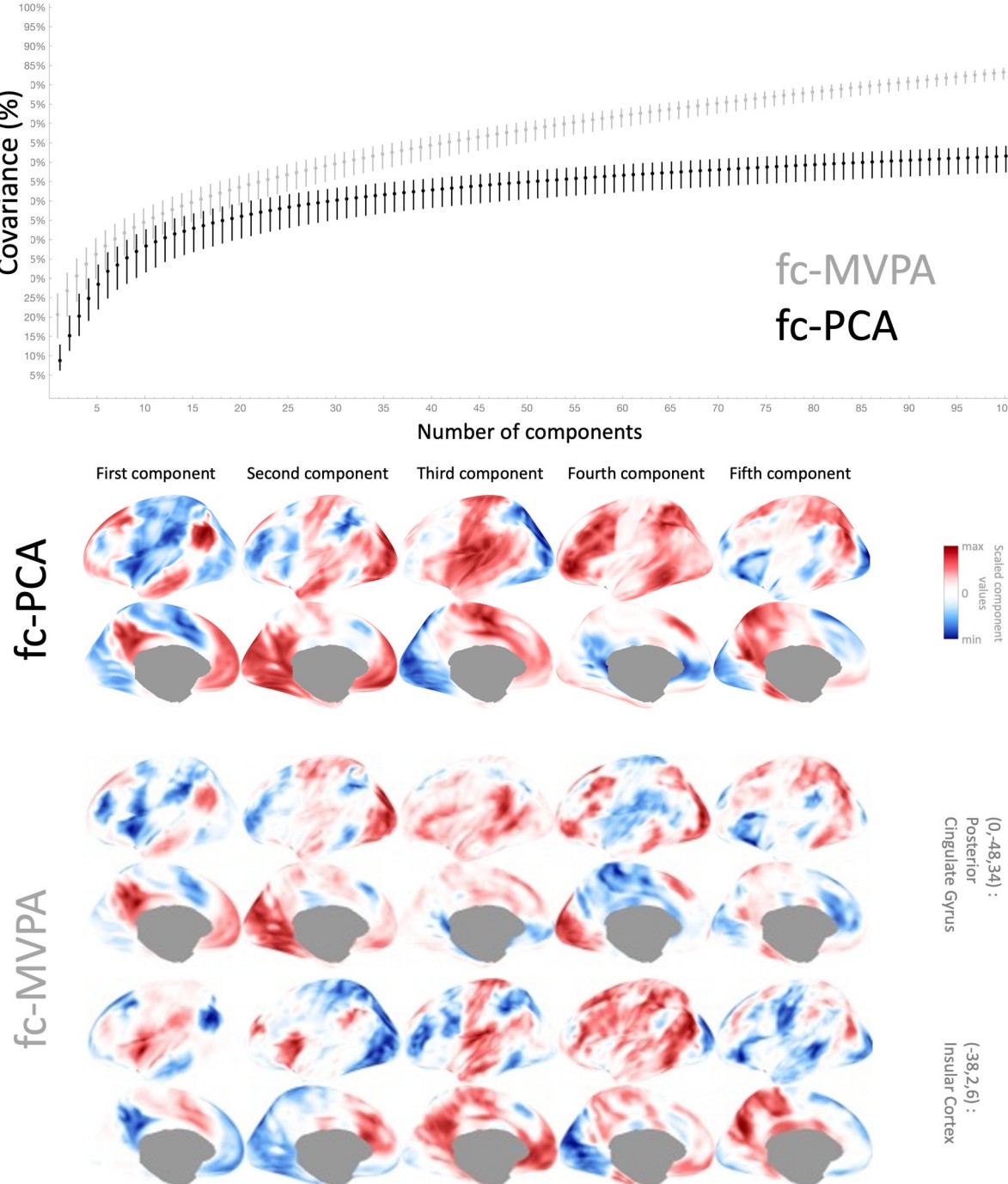

**Fig 6. Comparison between PCA and MVPA components. Top**: Median (dots) and 25%-75% percentile range (vertical lines) of the total covariance in functional connectivity patterns at each voxel explained cumulatively by the first *k* components from a functional connectivity Principal Component Analysis (black dots and lines), and by the first *k* fc-MVPA eigenpatterns (light gray dots and lines), from the analysis of the same sample dataset (Cambridge, n = 198 dataset). **Bottom**: First five principal components from PCA (first row) and from fc-MVPA (second and third row, first five eigenvariates shown only at two sample locations: posterior cingulate and anterior insula). Each row shows individual components sorted from first/left to fifth/right, projected to a left hemisphere lateral view (top image) and medial view (bottom image), on a relative color scale ranging from blue (highest negative values for each component) to yellow (highest positive values). Larger explanatory power of fc-MVPA components compared to PCA (shown on top figure) stems largely from the ability of fc-MVPA components to adapt to the specificity of the functional connectivity patterns at each individual location (as exemplified in the bottom figures by the differences and commonalities between the components describing posterior cingulate vs. anterior insula connectivity patterns).

Component or Independent Component Analyses of the same data). This, naturally, also supports the use of fc-MVPA eigenpattern scores in the context of brain-wide connectome inferences as a rich low-dimensional representation of the functional connectivity patterns for each subject.

## Simulations: validity and sensitivity of fc-MVPA statistics

In order to evaluate the validity and sensitivity of the general fc-MVPA inferential approach, we constructed a set of simplified simulations. All of the simulations consider a dataset with 50 subjects. Each subject's BOLD data encompass 50 timepoints and 1,000 voxels. For each voxel, the simulated BOLD timeseries contained a mixture of noise (independent samples from a Gaussian distribution for each timepoint, computed separately for each voxel and subject, and spatially convolved with a Gaussian filter with FWHM 10 voxels) and signal (independent samples from a Gaussian distribution for each timepoint, computed separately for each subject and shared across all voxels where the signal was present). The signal was only present in one half of the subjects and, among those subjects, only within 10% of all contiguous voxels. For each individual simulation Eq 5 was used to estimate the projection matrix P(x) at each individual voxel, and Eqs 7 & 8 were used to evaluate between-group differences in the patterns of connectivity between this voxel and all other voxels, using a 50-by-2 design matrix $G$ characterizing the two groups of subjects and a between-subjects contrast vector $C = [-1\ 1]$ evaluating the difference in functional connectivity between the two groups.

For each individual simulation, we computed the statistical parameter map $F(x)$ and the associated map of raw/uncorrected voxel-level p-values evaluating the null hypothesis separately for two different voxels: one where the signal was present (so the connectivity between that voxel and all of the other voxels is expected to differ between the two subject groups), and one where the signal was not present (so the connectivity between that voxel and all other voxels is not expected to differ between the two subject groups). We run 10,000 simulations, and from their results, we computed summary Receiver Operating Characteristic (ROC) curves describing the true positive rate (probability of a voxel showing a significant between-group difference in connectivity) as a function of different prescribed false positive rates (p-value threshold used to determine significance), for each of these two representative voxels. The results from the first voxel, where the signal was present, were used to obtain estimates of the sensitivity of voxel-level fc-MVPA connectome inferences (*sensitivity* analyses), and the results from the second voxel, where the signal was not present, were used to obtain estimates of the validity of this inferential procedure (*validity* analyses).

Each of the above sets of 10,000 simulations was repeated 50 times, each time using a different number of eigenpatterns retained (ranging between 1 and 50) in Eq 5. In addition, all of the above simulations were repeated under six different scenarios in order to further evaluate the robustness of the obtained sensitivity and validity estimates in the presence of: a) varying levels of spatial cross-correlation of BOLD noise (FWHM set to 1, and 25 voxels); b) varying numbers of timepoints in BOLD scanning sessions (10, and 100 samples); and c) varying numbers of subjects in the study (10, and 100 subjects).

The results of the *validity* analyses are shown in Fig 7. The reported voxel-level p-values (shown in the x-axis labeled as *false positive rate*) from fc-MVPA inferences matched very precisely the empirically observed false positive rates (shown in z-axis labeled as *positive rate*), with all tested conditions showing accurate diagonal ROC curves. Differences between reported voxel-level p- values and observed false positive rates were below ± 0.22% in 50% of all simulations, and below ± 0.98% in 99% of all simulations. When controlling voxel-level false positives at a 5% level, and across a total of 386 sets of different conditions evaluated, the

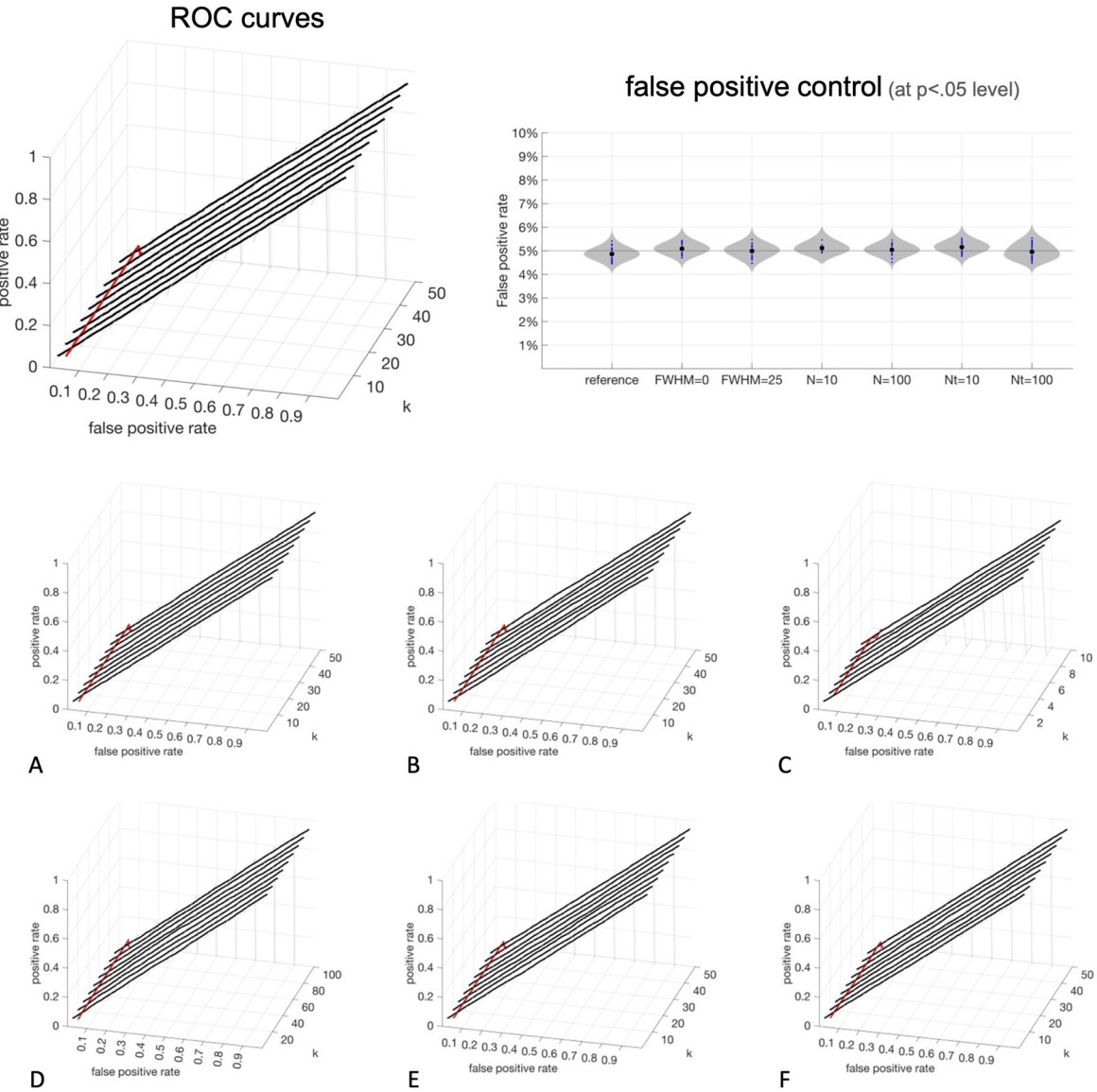

**Fig 7. Validation of fc-MVPA voxel-level inferences.** Analysis of Receiver Operating Characteristic curves evaluating between-group differences in functional connectivity under the null hypothesis (when there are no true differences in the population). **Top left**: surfaces, and highlighted thick black lines, show, for a chosen combination of false positive threshold (*false positive rate* x-axis) and number of eigenpatterns (*k* y-axis), the resulting proportion of false positive results (*positive rate* z-axis), where the fc-MVPA procedure would falsely conclude there is a significant difference in connectivity between the groups. The red line marks the observed rate of false positives when fixing the prescribed false positive rate threshold at a fixed 5% level (graphically, the intersection of each ROC surface and a vertical plane with constant false positive rate = 0.05), matching the expected 5% level. **Top Right**: Observed false positive rates (y-axis) when using fc-MVPA statistical analyses controlled at a p < .05 level across the reference simulations ('reference') and simulations evaluating different conditions (FWHM = 0, FHWM-25, N = 10, N = 100, Nt = 10, Nt = 100). The average (black dots) and histogram (gray surfaces) of the observed false positive rates across these simulations all indicate an appropriate match to the expected/prescribed false positive level (5%). **Bottom**: evaluating validity under different conditions: (**A**) low spatial autocorrelation (FWHM = 0); (**B**) large spatial autocorrelation (FWHM = 25 voxels); (**C**) low number of subjects (N = 10); (**D**) high number of subjects (N = 100); (**E**) short scanning session (Nt = 10); (**F**) long scanning session (Nt = 100).

empirical false positive rate observed across the 10,000 fc-MVPA analyses within each set ranged between 4.5% and 5.4% (Fig 7 top right). Statistics remained valid across the entire range of eigenpatterns tested up to the point where the number of eigenpatterns (*a* in Eq 8) equals the error degrees of freedom (*b* in Eq 8, equal to the number of subjects minus the number of

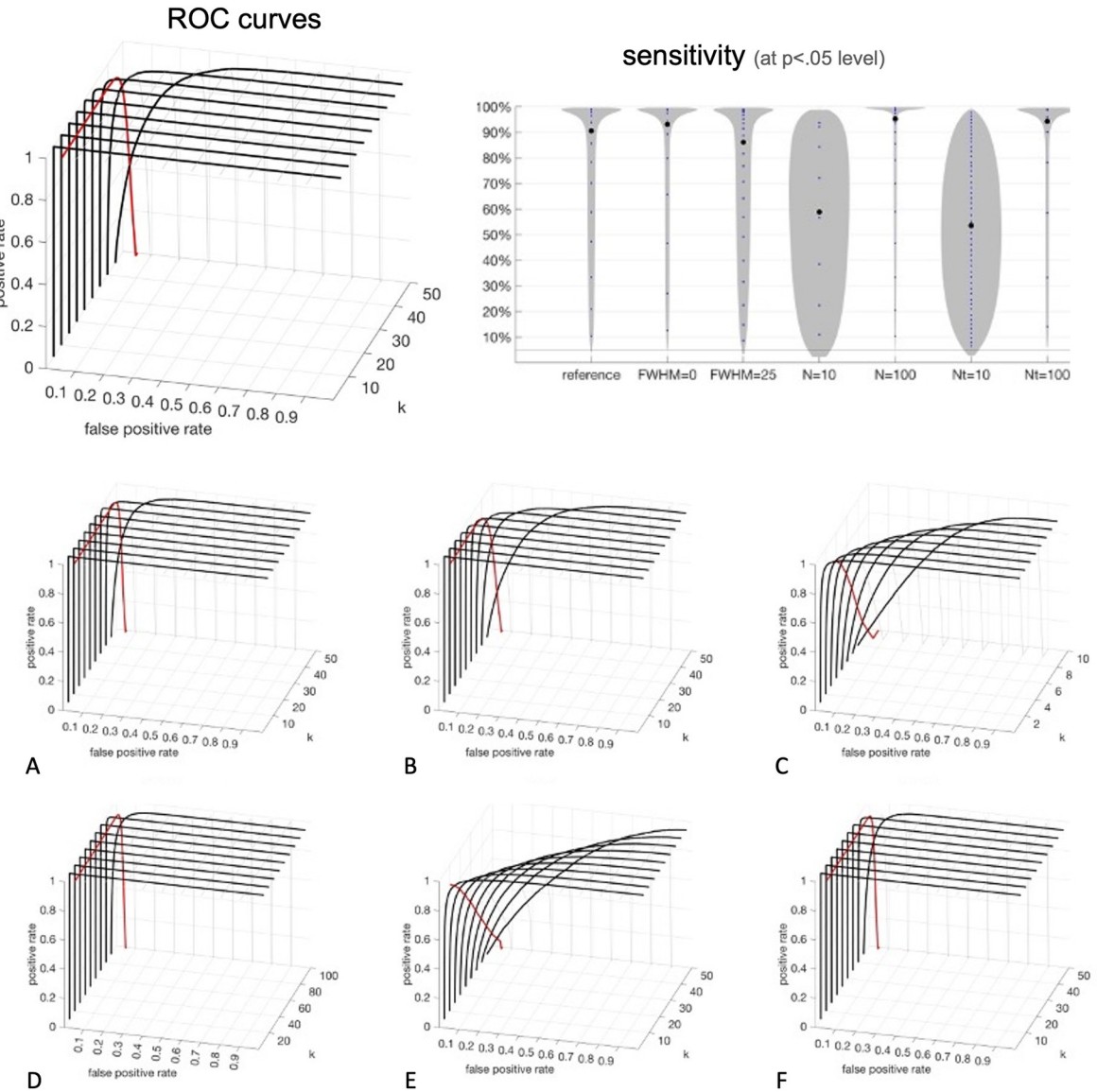

**Fig 8. Sensitivity of fc-MVPA voxel-level inferences.** Analysis of Receiver Operating Characteristic curves evaluating between-group differences in functional connectivity. **Top Left**: surfaces, and highlighted thick black lines, show, for a chosen combination of false positive threshold (*false positive rate* x-axis) and number of eigenpatterns (*k* y-axis), the resulting proportion of true positive results (*positive rate* z-axis), where the fc-MVPA procedure would correctly conclude there is a significant difference in connectivity between the groups in our reference simulations. **Top Right**: Observed true positive rates (y-axis) when using fc-MVPA statistical analyses controlled at a $p < .05$ level across the reference simulations ('reference') and simulations evaluating different conditions (FWHM = 0, FHWM = 25, N = 10, N = 100, Nt = 10, Nt = 100). The average (black dots) and histogram (gray surfaces) of the observed true positive rates, or proportion of significant results, across these simulations indicate that sensitivity is typically higher when using low or intermediate numbers of eigenpatterns, with poorer sensitivity when the number of timepoints for functional connectivity estimation is low (Nt = 10), or when the number of subjects included in the analysis is low (N = 10). **Bottom**: evaluating sensitivity under different conditions: (**A**) no spatial autocorrelation (FWHM = 0); (**B**) large spatial autocorrelation (FWHM = 25 voxels); (**C**) low number of subjects (N = 10); (**D**) high number of subjects (N = 100); (**E**) short scanning session (Nt = 10); (**F**) long scanning session (Nt = 100).

model regressors, or 48 in our simulations) where the data covariance becomes rank deficient and the likelihood ratio test assumptions no longer hold.

The results of the *sensitivity* analyses are shown in Fig 8. Generally, sensitivity was large across the entire range of eigenpatterns tested, only decreasing markedly as the number of

eigenpatterns approached their maximum possible value. For example, sensitivity at a $p < .05$ level was above 80% in the main simulations (with 50 subjects) as long as the number of eigen-patterns was kept below 42, below 92 in the simulations with 100 subjects, and below 4 in the simulations with 10 subjects. While optimal sensitivity will naturally vary on multiple factors, including the size and nature of the effects that we are trying to evaluate, several trends in sensitivity were apparent from different scenarios evaluated. In particular, the degree of spatial autocorrelation in the functional data (simulations A-B in Fig 8) appeared to have a relatively small effect on sensitivity, while the number of subjects in the study (simulations C-D) and the duration of the scanning session (simulations E-F) both had a larger impact. For example, when fixing the number of eigenvariates to 5, sensitivity to detect a group effect was above 99% at a $p < .05$ level in the simulations with 50 or 100 subjects, but sensitivity dropped to 56% when the number of subjects was only 10. Similarly, sensitivity at a $p < .05$ level was above 99% when the number of simulated timepoints (or equivalently, the number of effective degrees of freedom of the BOLD timeseries in a study after the denoising and bandpass filter-ing procedure) was above 50, but it dropped to 92% when the number of simulated timepoints was only 10.

## Conclusions

This manuscript presented the theory and motivation behind functional connectivity Multi-variate Pattern Analyses (fc-MVPA), both in the context of brain-wide connectome inferences, as well as a model-free characterization of the heterogeneity in functional connectivity across subjects. Fc-MVPA extends or complements other MVPA approaches commonly used in neu-roimaging in three different ways: first, to characterize a subject's mental state, instead of the patterns of activation surrounding each voxel considered by many MVPA applications, fc-MVPA considers the patterns of connectivity between each voxel and the rest of the brain; sec-ond, instead of a backward model focusing on decoding known properties of a subject or of the experimental paradigm, fc-MVPA uses a forward model focusing on testing a researcher's hypothesis about the subject's connectivity state across the entire connectome (brain-wide connectome inferences); and third, in addition to the above inferential framework, fc-MVPA also provides a model-free characterization of the sources of intersubject heterogeneity in con-nectivity patterns.

Monte Carlo simulations showed that fc-MVPA inferences remain valid for the entire range of evaluated scenarios, including using any arbitrary number of eigenpattern scores, dif-ferent sample sizes, and scanning session durations. Simulations and example analyses of gen-der-related differences in functional connectivity illustrated the high sensitivity of fc-MVPA inferential statistics to detect meaningful effects across the entire human connectome. In addi-tion, an example analysis of fc-MVPA eigenpatterns in functional connectivity during resting state showed rich and varied sources of intersubject heterogeneity in functional connectivity.

One of the main practical advantages of fc-MVPA in the context of brain-wide connectome inferences, is that it combines the benefits of pattern analysis techniques, such as the increased interpretability and reduced noise of lower-dimensional projections, with the benefits of a clas-sical statistical framework, such as the ability to use popular approaches to group-level analyses (e.g. GLM's ANOVA and regression framework), novel multiple comparison techniques (e.g. TFCE), and well understood statistical control procedures (e.g. ANCOVA in this manuscript example analyses). Similarly, fc-MVPA eigenpattern representations offer a natural way to extend common multidimensional reduction approaches in neuroimaging, such as ICA or PCA, to begin considering the specificity of these lower-dimensional representations across different brain areas. From its theoretical and practical advantages, we believe that fc-MVPA

can be a powerful and hopefully useful tool for researchers to further explore the complexities of the human connectome.

## Supporting information

**S1 Appendix. Efficient computation of multivariate patterns.**
(DOCX)

**S2 Appendix. Preprocessing and analysis of resting state functional data.**
(DOCX)

**S1 Table. Glossary of terms in manuscript equations.**
(DOCX)

## Author Contributions

**Conceptualization:** Alfonso Nieto-Castanon.

**Formal analysis:** Alfonso Nieto-Castanon.

**Methodology:** Alfonso Nieto-Castanon.

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
