## [Decision Letter · Decision Letter 0]

13 Aug 2022

Dear Dr Nieto-Castanon,

Thank you very much for submitting your manuscript "Brain-wide connectome inferences using functional connectivity MultiVariate Pattern Analyses (fc-MVPA)" for consideration at PLOS Computational Biology. As with all papers reviewed by the journal, your manuscript was reviewed by members of the editorial board and by several independent reviewers. The reviewers appreciated the attention to an important topic. Based on the reviews, we are likely to accept this manuscript for publication, providing that you modify the manuscript according to the review recommendations.

In particular it would be great if this method would be backed up by some code in great shape, with a nice entry-level notebook. This would ultimately increase the impact and usability of this work.

Sincerely,

Daniele Marinazzo

Deputy Editor

PLOS Computational Biology

Daniele Marinazzo

Deputy Editor

PLOS Computational Biology

[LINK]

Reviewer's Responses to Questions

**Comments to the Authors:**

Reviewer #1: The author has previously provided a new tool for performing MVPA for inference in the context of functional connectivity (fc-MVPA). Here, the author describes fc-MVPA in detail, including its formal definition and properties, an empirical example to demonstrate its use and interpretation, and a simulation to demonstrate its valid control of false positive rates and its sensitivity under various conditions. Overall, this work provides a valuable and comprehensive overview for understanding an important tool for functional connectivity analysis. The description of how to interpret various components (e.g., effect size reporting, eigenpatterns) is particularly valuable. I only had a few minor suggestions.

“Classical MVPA analyses (e.g. Norman et al. 2006) attempt to predict, from these or other searchlight patterns, properties of the subjects (e.g. patients vs. controls) or properties of the experimental paradigm (e.g. pre- vs. after- intervention). These are often referred to as backward models (Haufe et al. 2014), and typically focus on machine learning classification models. Unlike those forms of MVPA, fc-MVPA uses instead a forward model of the data, attempting to predict the shape of these searchlight patterns from subject and experimental paradigm information.” (p 7)

The use of “predict” here for both predictive and explanatory modeling could be a little confusing. Although “predict” is historically used for explanatory procedures (e.g., regression), they do not actually typically involve any prediction (e.g., of unseen data) and it is now becoming more accepted to reserve “predict” for machine learning contexts. Consider something like “estimate the relationship between these searchlight patterns” instead.

Can the author provide a reference to any similar formal descriptions available for traditional “backwards model” MVPA in the literature? I found myself wanting to consult an additional text to see whether there were analogues for some details of the presented procedure (e.g., choice of a spatial basis).

Recent work has shown that multivariate inferential methods have better power than (mass) univariate methods for functional connectivity. If the author wants, it may be helpful for empirically motivating the present work:

Noble, S., Mejia, M., Zalesky, A., & Scheinost, D. (2021). Leveling up: improving power in fMRI by moving beyond cluster-level inference. BioRxiv.

Reviewer #2: This manuscript presents foundational materials and empirical results to support the use of MVPA with BOLD low-frequency MRI data to allow for voxel-wise comparisons of functional connectivity differences using the standard general linear model (e.g., fc-MVPA). The code and software implementation of the proposed MVPA technique has been made widely available by the submitting author, and the technique, as implemented in the MATLAB CONN toolbox (see https://www.nitrc.org/projects/conn), has been used in no less than 26 peer-reviewed publications with moderate to high impact factor ratings.

This manuscript provides valuable clarity into the theoretical underpinnings involved in generating eigenpatterns from singular value decomposition procedures, which further reinforces the utility of the fc-MVPA technique. Additionally, the manuscript provides critical support for the technique through simulation analyses to determine the impact of eigenpattern selection on the validity and sensitivity of the technique with various sample sizes.

Overall, I found the study to be interesting and the technique to be a valuable addition to the neuroimaging community, as it potentially addresses multiple problems inherent to traditional parcellation-based neuroimaging techniques. I found very little to fault in the manuscript aside from a few grammatical mistakes, though I would have liked to have seen more discussion on how regression models using fc-MVPA might be cross-validated using the existing CONN toolbox or code. It may be beyond the scope of the manuscript to provide more end-user friendly information on the implementation of the fc-MVPA technique, but the authors are encouraged to provide this information as either supplementary materials or link to materials in the toolbox’s NITRC database listing.

Reviewer #3: Generally, I find that having an article detailing this work is a useful endevour and that a report on this is missing in the literature.

At the moment, the article is between a tutorial targetting researchers who will apply the method and a mathematical background information on the method developed targetting a more methodological population.

It is regretable that the documentation of the software package is not more deeply integrated to the description of the method, for instance with a Jupyter book/notebook.

I also found confusing the name of the method, as MVPA is a well established method in brain imaging analysis but does not rely on the same principles as fc-MVPA. This is bound to induce confusion in the minds and in the literature. I recommend that the method is renamed avoiding the term "pattern" for instance, or using MCOR as a core acronym.

The simulation section could be pushed to an appendix. I found the numbers chosen fairly extreme (eg FWHM either 0 or 100). Choices that span a more realistic fMRI analysis setting would be preferable.

Last, I think that leaving the method without a reasonable way to estimate the number of eigen components to select is very risky. Researchers will try several values and most likely only report the "most interesting" results - leading to some form of p-hacking. There are a number of possible methods to at least guide this choice, at least the danger of selecting a value after having seen the statistical results should be described and avoided.

Details:

Can the author better explain / comment on the following ?

* Eq. 1: \\epsilon(x,y) seems to be implementing heterocedasticity across subjects, is OLS still the best minimal variance estimator ?

* The index n seems to be indexing subjects in eq 1, but then in eq 2 the lenght of r_n(x,y) is the number of voxel y. It would be easier to just define the dimensions of these vectors and matrix. It seems that \\bold{r_n(x)} is of dimension 1,p where p is the number of pixels, but it would be clearer with explicit dimensions

* there are inconsistencies in what is a column and a row vector,

* some matrix definitions come several lines after their use, it is generally ok but in this case these definitions come fairly late

* Eq 5 would be easier to read as variances or squared norms of the corresponding matrices. Eg define the variance of the estimated effects and use this in Eq 5.

* Eq 8 doesnt seem to include the dimentionality reduction - ie the definition of H and W have not been updated - I may have missed something here.

* $\\Omega$ is not precisely defined ("each significant cluster Ω") - it is unclear why we have to take integrals rather than sum which would make the material more accessible

* Between subjects contrasts are not necessarily one dimensional - eg think of the F statistics for difference between 3 groups, where the relevant contrast would be an "F-contrast" (eg \\citep{Poline2007})

* Effect sizes are difficult to understand for a non methodological oriented reader. If the article is aiming at the researchers who will apply the method, the should be better explained. More importantly, since all data are used to the SVD dimension reduction, they will be inflated. An estimation of this inflation should at least be provided (eg by separating the dataset into train and test sets), ideally a way to estimate non biaised effect sizes or correct for the bias would be provided.

* I would try to relate the effect sizes to more interpretable values, for instance the correlation differences averaged across voxels, which would be a location dependent measure.

* the author should be careful with the choice of colormap. It is well documented why jet / similar colormap should be avoided.

* Fig 8: it would be easier to read to figure with a selection of say 3 or 4 number of components and show the ROC curves in 2D. At the moment, there are redundancies and also some of the curves are hard to see.

**Have the authors made all data and (if applicable) computational code underlying the findings in their manuscript fully available?**

Reviewer #1: Yes

Reviewer #2: Yes

Reviewer #3: Yes

PLOS authors have the option to publish the peer review history of their article (what does this mean?). If published, this will include your full peer review and any attached files.

Reviewer #1: No

Reviewer #2: No

Reviewer #3: No

Figure Files:

Data Requirements:

Reproducibility:

References:

---

## [Editor Report · Decision Letter 1]

29 Sep 2022

Dear Dr Nieto-Castanon,

Thank you very much for submitting your manuscript "Brain-wide connectome inferences using functional connectivity MultiVariate Pattern Analyses (fc-MVPA)" for consideration at PLOS Computational Biology. As discussed per email. Please note while forming your response, if your article is accepted, you may have the opportunity to make the peer review history publicly available. The record will include editor decision letters (with reviews) and your responses to reviewer comments. If eligible, we will contact you to opt in or out

Sincerely,

Daniele Marinazzo

Section Editor

PLOS Computational Biology

Daniele Marinazzo

Section Editor

PLOS Computational Biology

Figure Files:

Data Requirements:

Reproducibility:

References:

---

## [Editor Report · Decision Letter 2]

4 Oct 2022

Dear Dr Nieto-Castanon,

We are pleased to inform you that your manuscript 'Brain-wide connectome inferences using functional connectivity MultiVariate Pattern Analyses (fc-MVPA)' has been provisionally accepted for publication in PLOS Computational Biology.

Best regards,

Daniele Marinazzo

Section Editor

PLOS Computational Biology

Daniele Marinazzo

Section Editor

PLOS Computational Biology

---

## [Editor Report · Acceptance letter]

9 Nov 2022

PCOMPBIOL-D-22-00984R2 

Brain-wide connectome inferences using functional connectivity MultiVariate Pattern Analyses (fc-MVPA)

Dear Dr Nieto-Castanon,

I am pleased to inform you that your manuscript has been formally accepted for publication in PLOS Computational Biology. Your manuscript is now with our production department and you will be notified of the publication date in due course.

With kind regards,

Zsanett Szabo
